# Tumor Treating Fields Concomitant with Sorafenib in Advanced Hepatocellular Cancer: Results of the HEPANOVA Phase II Study

**DOI:** 10.3390/cancers14061568

**Published:** 2022-03-18

**Authors:** Eleni Gkika, Anca-Ligia Grosu, Teresa Macarulla Mercade, Antonio Cubillo Gracián, Thomas B. Brunner, Michael Schultheiß, Monika Pazgan-Simon, Thomas Seufferlein, Yann Touchefeu

**Affiliations:** 1Department for Radiation Oncology, University Medical Centre Freiburg, Robert-Koch-Straße 3, 79106 Freiburg, Germany; anca.grosu@uniklinik-freiburg.de; 2Department of Medical Oncology, Vall d’Hebrón University Hospital and Vall d’Hebrón Institute of Oncology, 08035 Barcelona, Spain; tmacarulla@vhebron.net; 3Department of Medical Oncology, Centro Integral Oncológico Clara Campal HM CIOCC, Hospital Universitario HM Sanchinarro, 28050 Madrid, Spain; acubillo@hmhospitales.com; 4Department of Radiotherapy, University Hospital Magdeburg, 39120 Magdeburg, Germany; thomas.brunner@med.ovgu.de; 5Department of Medicine II, University Hospital Freiburg, 79106 Freiberg, Germany; michael.schultheiss@uniklinik-freiburg.de; 6Department of Infectious Diseases and Hepatology, Wroclaw Medical University, 50-367 Wrocław, Poland; monikapazgansimon@gmail.com; 7Department of Internal Medicine, University Hospital Ulm, 89081 Ulm, Germany; thomas.seufferlein@uniklinik-ulm.de; 8Nantes Université, CHU Nantes, Institut des Maladies de l’Appareil Digestif (IMAD), Hépato-Gastroentérologie, Inserm CIC 1413, F-44000 Nantes, France; yann.touchefeu@chu-nantes.fr

**Keywords:** hepatocellular carcinoma (HCC), liver cancer, solid tumor, sorafenib, TTFields

## Abstract

**Simple Summary:**

Advanced hepatocellular carcinoma (HCC) is an aggressive liver cancer with limited treatment options and poor prognosis. Tumor-Treating Fields (TTFields) are electric fields that disrupt cancer cell division and are approved for glioblastoma and mesothelioma treatment. Laboratory research and clinical data from other solid tumor types provided a rationale to investigate whether TTFields combined with a standard treatment (sorafenib) was effective and well tolerated in advanced HCC, with the aim of ultimately improving treatment for these patients. Overall, 27 patients with large tumors and advanced disease were included in this study. Results showed that TTFields with sorafenib reduced the tumor size in 9.5% of patients compared with 4.5% in other studies examining sorafenib alone and was well tolerated. Reduction of tumor size was even better in patients who received TTFields for ≥12 weeks (18%). Results support further investigation of TTFields used with sorafenib in a larger phase III clinical study.

**Abstract:**

Advanced hepatocellular carcinoma (HCC) is an aggressive disease associated with poor prognosis. Tumor Treating Fields (TTFields) therapy is a non-invasive, loco-regional treatment approved for glioblastoma and malignant pleural mesothelioma. HCC preclinical and abdominal simulation data, together with clinical results in other solid tumors, provide a rationale for investigating TTFields with sorafenib in this patient population. HEPANOVA was a phase II, single arm, historical control study in adults with advanced HCC (NCT03606590). Patients received TTFields (150 kHz) for ≥18 h/day concomitant with sorafenib (400 mg BID). Imaging assessments occurred every 12 weeks until disease progression. The primary endpoint was the overall response rate (ORR). Safety was also evaluated. Patients (*n* = 27 enrolled; *n* = 21 evaluable) had a poor prognosis; >50% were Child–Turcotte–Pugh class B and >20% had a baseline Eastern Clinical Oncology Group performance status (ECOG PS) of 2. The ORR was higher, but not statistically significant, for TTFields/sorafenib vs. historical controls: 9.5% vs. 4.5% (*p* = 0.24), respectively; all responses were partial. Among patients (*n* = 11) with ≥12 weeks of TTFields/sorafenib, ORR was 18%. Common adverse events (AEs) were diarrhea (*n* = 15/27, 56%) and asthenia (*n* = 11/27, 40%). Overall, 19/27 (70%) patients had TTFields-related skin AEs; none were serious. TTFields/sorafenib improved response rates vs. historical controls in patients with advanced HCC, with no new safety concerns or related systemic toxicity.

## 1. Introduction

Hepatocellular carcinoma (HCC) is the most common form of primary liver cancer, accounting for approximately 90% of cases globally [1]. HCC has a poor prognosis, and historically, curative treatments have been limited to hepatic resection and transplant. Local ablation is the mainstay for unresectable, early-stage HCC, with trans-arterial chemoembolization (TACE) preferred for patients with Barcelona Clinic Liver Cancer (BCLC)-stage B/C tumors. Selective internal radiation therapy (SIRT) is an option for patients with BCLC-B/C tumors who are not eligible for TACE due to portal vein thrombosis [2,3]. The approval of systemic therapies such as tyrosine kinase inhibitors, anti-angiogenic inhibitors and immune checkpoint inhibitors has improved outcomes for patients with advanced HCC. For patients with advanced HCC not amenable for resection or local therapies, the recommended first-line therapy is atezolizumab (immune checkpoint inhibitor) plus bevacizumab (humanized monoclonal antibody against anti-vascular endothelial growth factor [VEGF]) [4], with lenvatinib (tyrosine kinase inhibitor) [5] or sorafenib (multi-kinase inhibitor) [6] given as alternatives if contraindications for immune therapies exist [2]. Sorafenib was the first systemic therapy to show an overall survival (OS) and progression-free survival (PFS) benefit in patients with unresectable advanced HCC [6,7]. Despite this, the prognosis for patients with advanced HCC is still poor and as such, an unmet need remains for this patient population [8]. Generally, systemic treatments are only recommended in patients with Child–Turcotte–Pugh (CTP) class A, with limited data and poor efficacy in CTP class B [2]. As such, many patients with advanced HCC are ineligible for systemic therapies due to high disease burden and severity. Given the nature of HCC, a combination of systemic treatment with a locally active and well-tolerated treatment may substantially improve patient outcomes.

Tumor-Treating Fields (TTFields) therapy is a non-invasive, loco-regional therapy that selectively disrupts division of cancer cells by delivering low-intensity, intermediate frequency, alternating electric fields to the tumor [9] via skin-placed arrays. TTFields exert their effect at an optimal frequency by causing mitotic arrest and apoptosis, ultimately slowing tumor growth and inducing cell death [9,10]. Specifically, TTFields disrupt the localization and orientation of highly polar molecules, such as tubulin and septin, and organelles within the cells by exerting electric forces. As a result, several key steps in mitosis are disrupted, including microtubule spindle assembly, localization of contractile elements around the cleavage furrow, chromosome segregation and cytokinesis [9,10,11]. Furthermore, emerging data suggest that, in addition to antimitotic effects, TTFields exert biophysical forces on a variety of charged and polarizable molecules to impact a number of biological processes, including DNA repair, autophagy, cell migration, permeability and immunological responses [12,13,14,15,16,17]. The multi-modal mechanisms of action of TTFields therapy are suggestive of broad applicability and combinatorial potential with other treatment options and modalities. Furthermore, as TTFields therapy involves externally generated fields delivered to the tumor, there is no active need to consider half-life [18], bioavailability, drug–drug interactions or other pharmacokinetic parameters, as with biologics (immunotherapy [19,20]) or chemotherapy [21]). Biological effects cease when the device is powered off. The physical-based mechanism of action further expands the combinatorial potential of TTFields.

TTFields therapy, which is generated by a portable, wearable medical device, may cause localized skin irritation underneath the arrays in some patients; however, this can be effectively controlled using topical therapies in most cases [22,23,24,25,26,27,28,29].

TTFields have shown encouraging preliminary efficacy and a tolerable safety profile in a number of pilot studies in a range of solid tumor types including pancreatic, ovarian and lung tumors when used concomitantly with systemic therapies [30,31,32,33]. In the approved indication of newly diagnosed glioblastoma (ndGBM), TTFields concomitant with temozolomide (TMZ) extended OS by 4.9 months and PFS by 2.7 months (vs. TMZ alone) [25,27]. Furthermore, in patients with malignant pleural mesothelioma (MPM), receiving TTFields and pemetrexed plus platinum-based chemotherapy (i.e., historic standard of care [SOC] prior to the recent Checkmate 743 study [34]), median OS was 18.2 months and median PFS was 7.6 months with TTFields therapy, which represents an improvement in expected survival outcomes [22]. Additionally, analysis of four phase I–II studies in pancreatic, ovarian and lung tumors revealed that TTFields therapy did not result in treatment-related pulmonary, cardiac, hematological or gastrointestinal toxicity [35]. The United States Food and Drug Administration has approved TTFields therapy for the treatment of adult patients with ndGBM and recurrent GBM [23,25,27,28] and for MPM [22,24]. TTFields are also CE-marked in the European Union [36], as well as in other countries and regions.

Previously reported early development preclinical and simulations data have shown that TTFields can be delivered effectively and safely to the liver [37]; in vitro TTFields led to a reduction in viability and clonogenic potential of human HCC cell lines at an optimal frequency of 150 kHz [38,39]. Furthermore, TTFields with sorafenib reduced cancer cell counts and enhanced apoptosis to a greater extent than either treatment alone. In vivo results confirmed in vitro data, demonstrating that TTFields monotherapy was effective against HCC cells and that concurrent use with sorafenib led to a further enhancement of efficacy. Specifically, tumor growth and volume were significantly reduced with TTFields and sorafenib compared with the control and either treatment alone [38,39]. Safety assessments showed no TTFields-related adverse events (AEs) associated with delivery to the abdomen of healthy rats [38,39]. Additional preclinical data revealed no changes in the safety parameters of healthy rats treated with TTFields (150 or 200 kHz) to the torso versus (vs). control animals [40]. Furthermore, simulation studies have demonstrated effectiveness and thermal safety of TTFields delivery to the human abdomen [41,42].

Preclinical data in HCC cell lines and in vivo models, together with abdominal simulations and clinical efficacy and safety data in other solid tumors, provide a rationale for investigating the efficacy and safety of TTFields with sorafenib in advanced HCC. Here, we report the results of a phase II study investigating the efficacy and safety of TTFields (150 kHz) therapy concomitant with sorafenib in patients with advanced HCC.

## 2. Materials and Methods

### 2.1. Study Design and Ethics Oversight

HEPANOVA was a prospective, open-label, phase II, single arm, historical control study designed to test the preliminary efficacy and safety of TTFields concomitant with sorafenib in adult patients with advanced HCC (NCT03606590); it was conducted at six sites across six European countries (Appendix A). The study design is shown in Figure 1. The protocol conformed to the ethical guidelines of the 1975 Declaration of Helsinki and was approved by the relevant Ethics Committee and Competent Authority at each participating site. The trial was conducted in compliance with good clinical practice guidelines (EN ISO 14155:2011) and national or regional regulations, as appropriate. All patients provided written informed consent.

All eligible adult patients were to receive TTFields (150 kHz) concomitant with sorafenib. Follow-up visits were conducted every four weeks, with computed tomography (CT) or magnetic resonance imaging (MRI) every 12 weeks until disease progression in the liver per RECIST version 1.0 [43]. Post-progression, patients were initially seen 30 days after discontinuation of TTFields, and then follow-up was performed every eight weeks by phone.

### 2.2. Patients

Adults (≥18 years of age) with HCC diagnosed by biopsy or by typical imaging criteria (CT/MRI) and alfa fetoprotein, BCLC stage 0–C, a CTP score of 5–8 points (corresponding to grade A–B8), an Eastern Cooperative Oncology Group performance status (ECOG PS) of 0–2 and life expectancy of ≥12 weeks were eligible for enrollment. Patients were excluded if they were candidates for surgical resection or local treatment (e.g., TACE, SIRT, radio-frequency thermal ablation, microwave ablation or surgery) or had concurrent or prior malignancy requiring anti-tumor treatment. Full inclusion and exclusion criteria are listed in the Appendix A.

### 2.3. Treatments

Continuous TTFields (150 kHz) therapy was generated using the NovoTTF-100L(P) system (Figure 2) and was delivered for ≥18 h/day through abdomen-placed arrays (Figure 2). Arrays remain in place during treatment; however, they can be removed during treatment breaks and during routine array changes. Furthermore, patients can carry or wear the field generator and battery in a bag connected to the arrays whilst receiving treatment as these components are essential for the generation of TTFields. TTFields therapy was initiated within seven days of enrollment and ±seven days from first sorafenib dose. Sorafenib was administered at a dose of 400 mg twice daily. Sorafenib treatment was continued until the patient was no longer clinically benefiting from therapy or until occurrence of toxicity attributed to sorafenib (as determined by each investigator). Both treatments were stopped following disease progression in the liver (RECIST version 1.0), occurrence of unacceptable toxicity or patient withdrawal.

### 2.4. Assessments and Outcomes

The primary endpoint was the overall response rate (ORR) compared with historical controls [6,44,45,46]. ORR was defined as the percentage of patients who experienced a complete or partial response (RECIST version 1.0 for HCC). Secondary efficacy endpoints were the in-field control rate at one year (percentage of patients who did not have progression confined to the right hypochondriac and epigastric anatomical regions at one year following enrollment), distant metastases-free survival rate at one year (percentage of patients who did not have new metastases outside the liver at one year), OS at one year and PFS at six and 12 months. Exploratory analyses of the disease control rate (DCR) and time-to-progression (TTP) were not pre-specified in the clinical investigation plan. The ORR, in-field control rate, PFS and TTP were all based on RECIST version 1.0 for HCC.

AEs were assessed and graded according to Medical Dictionary for Regulatory Activities (MedDRA) version 21.0 and Common Terminology Criteria for Adverse Events (CTCAE) version 4.0 for all patients who received ≥1 day of TTFields therapy. TTFields-related skin AEs were graded using modified criteria.

### 2.5. Statistical Analysis

The primary endpoint was estimated using an exact binomial distribution together with the 95% confidence interval (CI). The difference between the study population compared to the historical control was tested using a chi square test at a one-sided significance level of 0.05, with the denominator being the number of patients in the trial evaluable for response assessment (i.e., patients with ≥1 CT or MRI scan after baseline).

The assumption that sorafenib monotherapy would result in an ORR of 4.5% was based on historical data from studies in advanced HCC (Appendix A) [6,44,45,46]. The study was powered to detect an ORR rate of 20% in patients treated with TTFields compared to the 4.5% ORR calculated from historical control studies. A sample size of 25 patients was required to achieve a power of 77% at a one-sided alpha level of 0.05 using the one-sample exact test for proportion.

Secondary endpoints were presented descriptively; no formal hypothesis testing was conducted to avoid type I error duplicity. The in-field control rate was calculated one year after enrollment. Patients lost to follow-up before one year were not included in this analysis. Distant metastases-free survival at one year, 1-year OS and 1-year PFS rates were estimated for the intent-to-treat (ITT) population. Exploratory analyses (DCR and TTP) were conducted on data from the ITT population. Additional analyses of response (excluding TTP) were also performed in a subgroup of patients who completed ≥12 weeks of TTFields therapy.

AEs were collected during the period immediately following enrollment until 30 days after the cessation of treatment. AEs were summarized by their MedDRA version 21.0 preferred term within the system organ class by severity and were presented as incidence. Severe AEs were defined as grade 3 events according to CTCAE version 4.0. Serious AEs were defined as an AE that led to death or a serious deterioration in health or led to fetal distress, fetal death or a congenital abnormality or birth defect. Patients reporting multiple episodes of the same AE (i.e., same preferred term) were counted once.

## 3. Results

### 3.1. Patients

Of the 35 total patients screened for eligibility, 27 patients with HCC were enrolled and received TTFields (150 kHz) therapy (ITT population). One patient in the ITT population did not receive concomitant sorafenib treatment (Figure 3). Six patients (22%; all with a CTP score of 7–8 [i.e., class B]) died before the first follow-up scan at 12 weeks; therefore, imaging data were only available for the response analysis for 21 patients. Baseline characteristics of the patient population are summarized in Table 1. The median age was 65 years, and the majority of patients were male. Patients typically had a poor prognosis, with 51.8% of patients classified as CTP class B and 22.2% with an ECOG PS of 2 at baseline. In total, 14 (51.9%) patients had extrahepatic spread at baseline. The median (range) of prior treatments was 1 (0–6), and the time from diagnosis to enrollment was 25.6 weeks (1.9–345.9).

### 3.2. Treatments

The median (range) duration of treatment for TTFields and sorafenib was 10.0 weeks (0.3–73.9) and 17.7 (0.0–80.1) weeks, respectively. The mean TTFields usage time was 64% (standard deviation: 23%). Of the 27 patients included in the ITT population, 11 (41%) patients received ≥12 weeks of treatment with TTFields; five patients received ≥six months of treatment with TTFields.

### 3.3. Efficacy

The ORR (primary endpoint) was numerically higher by approximately twofold but was not significantly different for TTFields concomitant with sorafenib vs. historical control for sorafenib monotherapy: 9.5% vs. 4.5%, respectively; all responses were partial (Table 2). Among the subgroup of patients who received ≥12 weeks of TTFields concomitant with sorafenib treatment, ORR was almost four times that of the historical controls, 18.0% vs. 4.5%, respectively (Table 2). Therefore, the ORR in patients who received an adequate duration of TTFields therapy as specified in the study protocol was almost, but just less than, the targeted ORR (20%).

The in-field control rate at one year was 9.5% (Table 2) and the 1-year OS and PFS rates were 30% (95% CI, 11–52) and 23% (95% CI, 7–45), respectively, for the ITT population (Table 3). For patients who received ≥12 weeks of treatment, the in-field control rate at one year was 9.1% (Table 2) and the 1-year OS and PFS rates were 64% (95% CI, 30–85) and 28% (95% CI, 5–58), respectively. The distant metastases-free survival rate at one year in the ITT population based on Kaplan–Meier curve estimation was 26% (95% CI, 8–49), as shown in Table 3. For patients who received ≥12 weeks treatment, this increased to 30.5% (95% CI, 5–62).

Exploratory analyses reported a DCR of 76% in evaluable patients (*n* = 21; Table 2) and a median TTP of 8.9 months (95% CI, 3.1–not reached) in the ITT population (Table 3). The DCR was 91% among patients who received ≥ 12 weeks of treatment (*n* = 11), as shown in Table 2.

ORR was 9.1% for patients classified as CTP class A and 10.0% for those classified as CTP class B (Appendix A). One-year OS rates were 29% (95% CI, 17–68) and 27% (95% CI, 7–51), for classes A and B, respectively, while 1-year PFS and distant metastases-free survival rates were 29 (95% CI, 5–61) and 20 (95% CI, 3–47) and 38 (95% CI, 5–72) and 20 (95% CI, 3–47), respectively (Appendix A).

### 3.4. Safety

In total, 26 patients (96%) experienced ≥1 AE; the most frequent AEs were diarrhea (*n* = 15, 56%), asthenia (*n* = 11, 41%), decreased appetite (*n* = 8, 30%) and ascites (*n* = 6, 22%) (Table 4).

There were nine patients (33%) who reported ≥1 mild–moderate (grade 1–2) AEs, and the most frequently reported AEs were diarrhea (48%) and asthenia (33%). In addition, 16 patients (59%) had severe (grade 3–4) AEs. The most common was decreased appetite, reported by three patients (11%), as shown in Table 5. Of the 13 patients (48%) who experienced ≥1 serious AE during the study, all but one (pathological fracture) were observed in only one patient.

Of the 26 patients with ≥1 AE, seven (26%) experienced events were deemed not related to TTFields therapy. There were 19 patients (70%) with TTFields-related skin AEs (Appendix A): 18 patients (67%) had grade 1–2 events, and one patient (4%) had a grade 3 event of skin erosion. The most common grade 1–2 treatment-related skin AE was dermatitis, which was reported in five patients (19%). There were no observed device deficiencies that could have led to a serious adverse device effect. No serious AEs related to TTFields therapy were reported. There were 18 reported deaths as of 20 April 2021; none were deemed related to the study treatment.

## 4. Discussion

Despite advances in systemic therapies for advanced HCC, an unmet need remains to improve patient outcomes. This phase II study investigated the efficacy and safety of TTFields therapy concomitant with sorafenib in patients with advanced HCC. Data presented here demonstrate that TTFields (150 kHz) concomitant with sorafenib resulted in numerical (but not statistically significant) improvement in outcomes in patients with advanced HCC as compared to historical controls [6,44,45,46] and without an increase in systemic toxicity.

While the 9.5% ORR for TTFields-treated patients was below the expected figure of 20% and the result thus not statistically significant, the observed results are numerically better than those for historical controls and represent a marked clinical improvement. This improvement was observed despite the HEPANOVA population having a worse prognosis than that of the populations of the control studies. In HEPANOVA, 52% of patients with advanced HCC had a baseline CTP score of ≥7, compared with ≤5% in two of the historical sorafenib studies [6,45] and 28% in another sorafenib study [44]. Furthermore, 22% of the HEPANOVA population had a reported baseline ECOG PS of two, compared with ≤15% reported in other studies in which patients with advanced HCC and an ECOG PS of 2 had been included and received treatment with sorafenib monotherapy [6,47]. In the HEPANOVA study, 30% of patients survived to one year and the median TTP was 8.9 months. Although the 1-year OS rates reported here are lower than the 44% previously reported by other sorafenib studies, [6,48] the HEPANOVA population experienced longer TTP than the 4.0–5.5 months reported by other studies [6,45]. It should be noted that direct comparisons cannot be made between these studies given the differences in trial design and study populations [6,45,48]. The lower 1-year survival rates reported here can be explained by the poor prognosis of the HEPANOVA patient population, with six patients dying before 12 weeks. These findings should not diminish the improvements in ORR reported here; rather, they are even more remarkable given the poor prognosis of the patients in HEPANOVA who were treated with TTFields concomitant with sorafenib.

Studies of real-world data for comparable treatment regimens of sorafenib alone observed DCRs of 23.6–47.0%, much lower than the 76% reported here with TTFields concomitant with sorafenib, despite the HEPANOVA patient population having a worse prognosis [47,48,49].

The mean TTFields usage time was 64%, which is lower than the 75% recommended per protocol. The results of a subgroup analysis of the phase III EF-14 trial (i.e., TTFields concomitant with maintenance TMZ vs. TMZ monotherapy in ndGBM) showed that increased usage of TTFields is a positive prognostic factor [50]. As such, if TTFields usage can be increased, potential exists for further improving efficacy above outcomes reported here. The lower than recommended usage time may be partially explained by the very poor prognosis of the population. Additionally, the first-generation model of TTFields was used in the present study. The second-generation model, which is half the size and weight, may offer increased usability and is now available for future studies. Moreover, this was the first TTFields study in patients with HCC and, as such, the site personnel gained preliminary experiences and learning that will be implemented in future studies. Another important consideration is the death of six patients within 12 weeks of enrollment, before per protocol imaging assessments could be performed. As a result, any early response that these patients may have experienced is not reflected in these data.

There were no systemic toxicities reported with TTFields, other than those expected for sorafenib [51]. The lack of additional systemic AEs is important, given that many combination therapies either approved for, or being investigated for use in advanced HCC, are associated with a myriad of systemic AEs [52], which may not be tolerated by patients with advanced HCC given the burden of disease. TTFields-associated AEs were in line with those reported in previous studies, with manageable/resoluble skin AEs localized beneath arrays being the most commonly reported [22,23,24,25,26,27,28,29]. Most skin AEs were low grade and non-serious (Appendix A); only one patient (4%) experienced a TTFields-associated grade 3 AE. Such AEs are likely related to the chronic exposure of the skin to potential array irritants and allergens that are required for application of TTFields (e.g., hydrogel, medical adhesives) [53]. The risk of skin irritation can be minimized by regularly changing arrays, changing array layout with regular array shifts (1–2 cm), as well as careful hygienic removal and proper ventilation of arrays [53]. Skin AEs like those reported here are manageable and perhaps preventable with proper practical and proactive symptom-based skincare management measures [54]. Furthermore, torso safety data previously reported with the use of TTFields in patients with MPM [22], as well as other solid tumor studies of TTFields [30,32,33], have not highlighted additional safety signals, thereby supporting the use of TTFields in the abdominal region.

Study limitations include the small patient population, the single-arm design and the lack of randomization, which although standard for a phase II study, limits the generalizability of the findings. However, future clinical investigation in a phase III study will aim to further explore the efficacy and safety of TTFields concomitant with SOC systemic therapy in a larger patient population with advanced HCC. Furthermore, the poor prognosis of the population (i.e., >50% of patients classed as CTP class B and >20% with an ECOG PS of 2 at baseline) translated to a short duration of treatment with TTFields and a substantial proportion of patients dying before ORR assessments could be carried out. As the study was powered to evaluate 25 patients, such loss of patients early in the study may have impacted the strength of the efficacy findings reported here. Additionally, some patients may have received other HCC treatments, such as immunotherapy, as part of clinical studies conducted prior to the current trial. These factors, particularly the loss of heavily burdened patients early in the study, shall be considered when designing future studies.

## 5. Conclusions

TTFields (150 kHz) therapy concomitant with sorafenib resulted in numerically improved response rates (~2-fold increase) versus sorafenib monotherapy in adult patients with unresectable advanced HCC, with no new safety signals/concerns or related systemic toxicity identified with the addition of TTFields. Furthermore, the response rate improved in patients who received ≥12 weeks of TTFields concomitant with sorafenib. Taken together, and in addition to the preclinical/simulations data, these clinical data suggest that TTFields concomitant with sorafenib is a feasible and tolerable therapeutic option. Given the potential for added benefit with TTFields in this high-risk patient population with unmet needs, the concurrent use of TTFields with current SOC treatment warrants further investigation in a larger, randomized, phase III clinical study. Overall, data suggest that continuous TTFields therapy is tolerable and efficacious in HCC with potential for broad application, as evidenced by clinical efficacy in GBM and MPM.

## Figures and Tables

**Figure 1 cancers-14-01568-f001:**
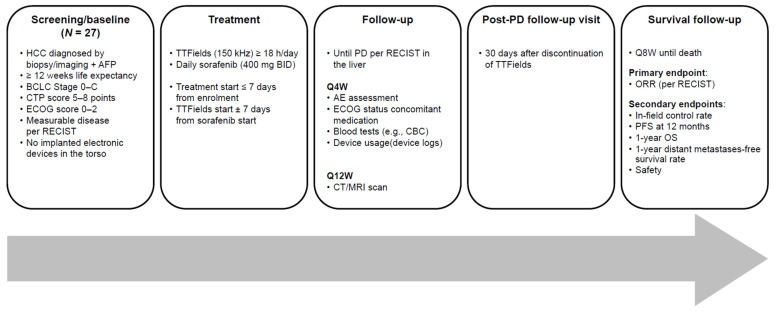
Study design. AE = adverse event; AFP = alfa fetoprotein; BCLC = Barcelona clinic liver cancer staging; BID = twice daily; CBC = complete blood count; CTP = Child–Turcotte–Pugh; CT = computed tomography; ECOG = Eastern Cooperative Oncology Group; HCC = hepatocellular carcinoma; MRI = magnetic resonance imaging; ORR = overall response rate; OS = overall survival; PD = progressive disease; PFS = progression-free survival; Q4W = every four weeks; Q8W = every eight weeks; Q12W = every 12 weeks; RECIST = response evaluation criteria in solid tumors; TTFields = Tumor-Treating Fields.

**Figure 2 cancers-14-01568-f002:**
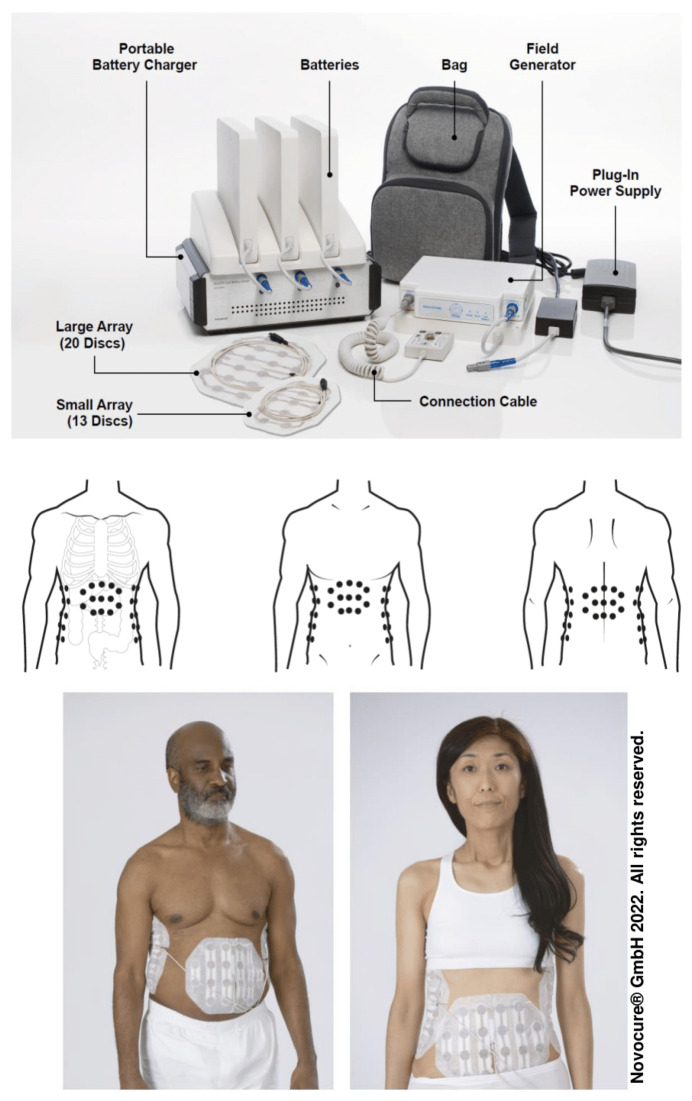
Models shown wearing arrays; models are actors and not patients. Reused with permission from © 2022 Novocure GmbH-all rights reserved.

**Figure 3 cancers-14-01568-f003:**
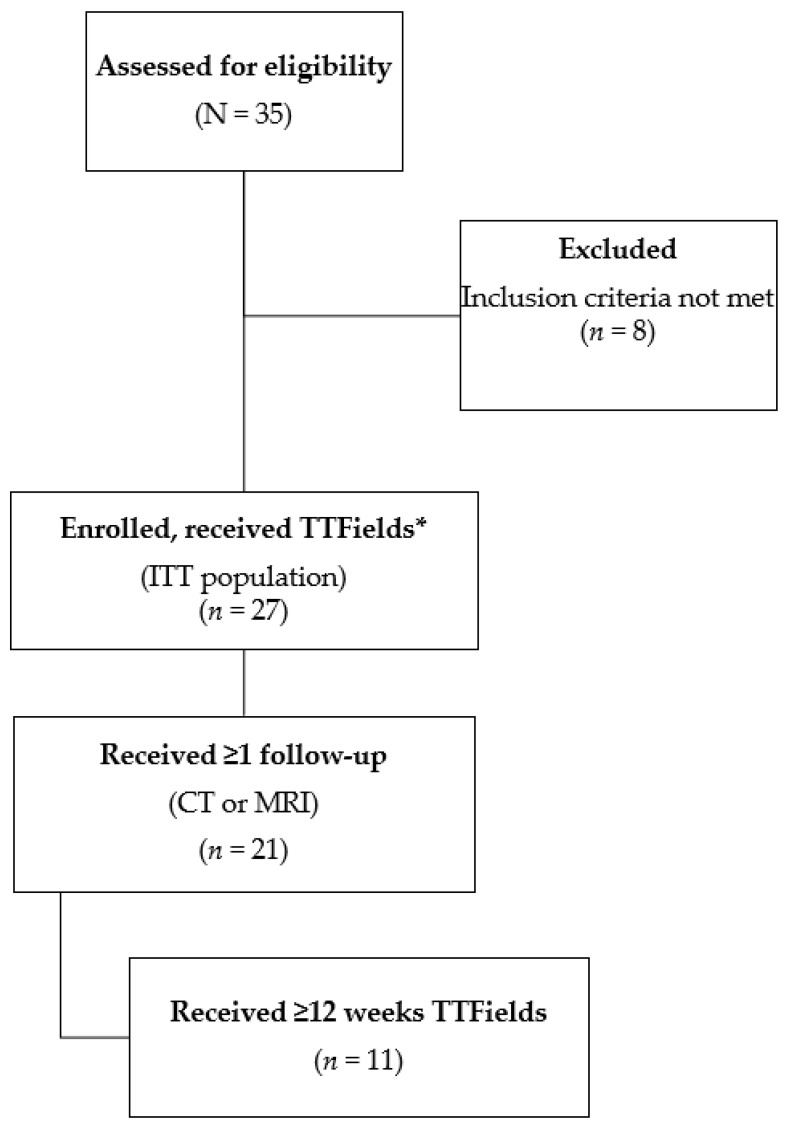
Patient flow diagram. * One patient did not receive concomitant sorafenib treatment.

**Table 1 cancers-14-01568-t001:** Patient baseline and disease characteristics treated with TTFields concomitant with sorafenib.

Characteristics	TTFields + Sorafenib (*n* = 27)
Age, years, median (range)	65 (28–85)
Sex, *n* (%)	
Female	1 (3.7)
Male	26 (96.3)
ECOG performance status, *n* (%)	
0	12 (44.4)
1	9 (33.3)
2	6 (22.2)
CTP score	
5	9 (33.3)
6	4 (14.8)
7	10 (37.0)
8	4 (14.8)
Number of prior treatments, median (range)	1 (0–6)
BCLC stage, *n* (%)	
0	1 (3.7)
B	5 (18.5)
C	21 (77.8)
Etiology, *n* (%)	
HBV	2 (7.4)
HCV	6 (22.2)
Alcoholic liver disease	9 (33.3)
Non-alcoholic fatty liver	3 (11.1)
Alcohol and dysmetabolism	1 (3.7)
Alcoholic liver disease and non-alcoholic fatty liver	1 (3.7)
NASH	1 (3.7)
Cirrhosis	1 (3.7)
Other	1 (3.7)
Missing	2 (7.4)
Extrahepatic spread, *n* (%)	14 (51.9)
Time from diagnosis to enrollment, median (range) weeks	25.6 (1.9–345.9)
Alpha-fetoprotein, median (range) ng/mL *	80.6 (1.0–4.7 × 10^6^)

* Data available for 26 patients. BCLC = Barcelona Clinic Liver Cancer, CTP = Child–Turcotte–Pugh; ECOG = Eastern Cooperative Oncology Group; HBV = hepatitis B virus, HCV = hepatitis C virus, NASH = non-alcoholic steatohepatitis; TTFields = Tumor-Treating Fields. Percentage values for patient subsets may not equal 100% due to rounding to the nearest integer.

**Table 2 cancers-14-01568-t002:** Response in all evaluable patients treated with TTFields (150 kHz) concomitant with sorafenib.

Outcome	TTFields + Sorafenib (*n* = 21)	TTFields ≥12 Weeks Usage + Sorafenib (*n* = 11)	Historical Control ^†^
Overall response rate, %	9.5	18	4.5 (*p* = 0.24) *
Level of response rate, %			
Complete	0	0	-
Partial	9.5	18	-
Stable disease	66.5	73	-
Disease control rate, %	76	91	-
In-field control rate at 1 year, %	9.5	9.1	–

* All evaluable patients vs. historical control. ^†^ Appendix A. TTFields = Tumor-Treating Fields.

**Table 3 cancers-14-01568-t003:** Time to event outcomes with TTFields (150 kHz) concomitant with sorafenib.

Outcome	TTFields + Sorafenib (*n* = 27)	TTFields ≥12 Weeks + Sorafenib (*n* = 11)
OS rate at 1 year, % (95% CI)	30 (11–52)	64 (30–85)
PFS rate at 12 months, % (95% CI)	23 (7–45)	28 (5–58)
Distant metastases-free survival rate at 1 year, % (95% CI)	26 (8–49)	30.5 (5–62)
Median time to progression, months (95% CI)	8.9 (3.1–not reached)	8.9 (5.8–not reached)

CI = confidence interval; OS = overall survival; PFS = progression-free survival; TTFields = Tumor-Treating Fields.

**Table 4 cancers-14-01568-t004:** All AEs occurring in >10% of patients treated with TTFields concomitant with sorafenib.

Preferred Term, *n* (%)	TTFields + Sorafenib (*n* = 27)
Patients with any ≥1 AE	26 (96)
Diarrhea	15 (56)
Asthenia	11 (41)
Decreased appetite	8 (30)
Ascites	6 (22)
Dermatitis	5 (19)
Dyspnea	5 (19)
Edema peripheral	5 (19)
Alanine aminotransferase increased	4 (15)
Palmar-plantar erythrodysesthesia syndrome	4 (15)
Skin erosion	4 (15)
Anemia	3 (11)
Aspartate aminotransferase increased	3 (11)
Constipation	3 (11)
Dry mouth	3 (11)
Hypertension	3 (11)
Nausea	3 (11)
Transaminases increased	3 (11)

AEs = adverse events; TTFields = Tumor-Treating Fields. Numbers and percentages have been rounded to the nearest integer.

**Table 5 cancers-14-01568-t005:** Severe (grade 3–4) AEs occurring in >5% of patients treated with TTFields (150 kHz) concomitant with sorafenib.

MedDRA Version 21.0 Preferred Term, *n* (%)	TTFields + Sorafenib (*n* = 27)
	Severe (Grade 3–4)
Patients with ≥1 any AE	16 (59)
Decreased appetite	3 (11)
Ascites	2 (7)
Diarrhea	2 (7)
Asthenia	2 (7)
Edema peripheral	2 (7)
Pathological fracture	2 (7)
Dyspnea	2 (7)
Palmar-plantar erythrodysesthesia syndrome	2 (7)
Hypertension	2 (7)

AE = adverse event; MedDRA = Medical Dictionary for Regulatory Activities; TTFields = Tumor-Treating Fields. Percentage values have been rounded to the nearest integer.

## Data Availability

The data generated and/or analyzed during the current study are available three years after date-of-publication, upon reasonable request. Please contact Uri Weinberg, Chief Scientific Officer, Novocure (weinberg@novocure.com) to request access.

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
