# Peer review of "Tumor Treating Fields Concomitant with Sorafenib in Advanced Hepatocellular Cancer: Results of the HEPANOVA Phase II Study"

_cancers, 2022, doi:10.3390/cancers14061568_

Round 1

Reviewer 1 Report

The authors provide an elegant and challenging Phase II clinical trial using TTFields for patients with advanced hepatocellular carcinoma. The study design is appropriate and the authors addressed the challenges of Phase II studies with a population that has an inherent high mortality. Overall I feel that the study is provocative and provides insightful information at this level of discovery. The limitation of the study is the impact of patient loss early in the study. With nearly 25% of patients falling out of the study prior to 12 weeks of therapy the authors must be cautious in over interpreting their results. The study was powered to evaluate 25 patients and with only 21 patients moving through analysis there is a data skew the will overweight a positive impact. Since this is a Phase II study I would classify this as a point that the authors need to consider when they move towards a Phase III study.

Author Response

Manuscript ID: 1602409

Manuscript title: Tumor Treating Fields Concomitant with Sorafenib in Ad-vanced Hepatocellular Cancer: Results of the HEPANOVA Phase II Study

Manuscript authors: Eleni Gkika, Anca-Ligia Grosu, Teresa Macarulla Mercade, Antonio Cubillo Gracián, Thomas B. Brunner, Michael Schultheiß, Monika Pazgan-Simon, Thomas Seufferlein and Yann Touchefeu

Please note that line numbers to signify the location of changes will be added to this document once the content is final, prior to resubmission

Response to Reviewer 1 Comments

Point 1: The authors provide an elegant and challenging Phase II clinical trial using TTFields for patients with advanced hepatocellular carcinoma. The study design is appropriate and the authors addressed the challenges of Phase II studies with a population that has an inherent high mortality. Overall I feel that the study is provocative and provides insightful information at this level of discovery. The limitation of the study is the impact of patient loss early in the study. With nearly 25% of patients falling out of the study prior to 12 weeks of therapy the authors must be cautious in over interpreting their results. The study was powered to evaluate 25 patients and with only 21 patients moving through analysis there is a data skew the will overweight a positive impact. Since this is a Phase II study I would classify this as a point that the authors need to consider when they move towards a Phase III study.

 Response 1: The authors thank the reviewer for their comment. We have included a line in the discussion noting that loss of patients early in the study should be taken into account in future studies.

Reviewer 2 Report

The manuscript by Gkika et al entitled “Tumor Treating Fields Concomitant with Sorafenib in Advanced Hepatocellular Cancer: Results of the HEPANOVA Phase II Study” describes that TTFields with sorafenib reduces tumor size in HCC patients compared to sorafenib alone and is well tolerated. Although the manuscript is written well, it has some shortcomings as follows:

Specific comments

  1. Some more detail about TTFields like type of generated field, mechanism of action against cancer cells etc.
  2. Patient detail like exclusion and inclusion criteria need to be included.
  3. It is not clear whether all parts, as shown in figure 1a, will be remain attached to the patients all the time ?
  4. Whether the pharmacological parameters that can be altered by TTFields are tested?

Author Response

Manuscript ID: 1602409

Manuscript title: Tumor Treating Fields Concomitant with Sorafenib in Ad-vanced Hepatocellular Cancer: Results of the HEPANOVA Phase II Study

Manuscript authors: Eleni Gkika, Anca-Ligia Grosu, Teresa Macarulla Mercade, Antonio Cubillo Gracián, Thomas B. Brunner, Michael Schultheiß, Monika Pazgan-Simon, Thomas Seufferlein and Yann Touchefeu

Please note that line numbers to signify the location of changes will be added to this document once the content is final, prior to resubmission

Response to Reviewer 2 Comments

Point 1: The manuscript by Gkika et al entitled “Tumor Treating Fields Concomitant with Sorafenib in Advanced Hepatocellular Cancer: Results of the HEPANOVA Phase II Study” describes that TTFields with sorafenib reduces tumor size in HCC patients compared to sorafenib alone and is well tolerated. Although the manuscript is written well, it has some shortcomings as follows

Response: The authors thank the reviewer for their comment and suggestions to improve the manuscript.

Point 2: Some more detail about TTFields like type of generated field, mechanism of action against cancer cells etc.

Response: Additional information on TTFields mechanism of action has been included in the introduction.

Point 3: Patient detail like exclusion and inclusion criteria need to be included.

Response: Please note that this information is provided in the supplementary information.

 Point 4: It is not clear whether all parts, as shown in figure 1a, will be remain attached to the patients all the time?

Response: We have added additional information in section 2.3 to address this point.

Point 5: Whether the pharmacological parameters that can be altered by TTFields are tested?

Response: Additional information has been included in the introduction section.

Reviewer 3 Report

This is an interesting manuscript describing HEPANOVA study, which was was a prospective, open-label, phase II, single arm, historical control study designed to test the preliminary efficacy and safety of TTFields concomitant with sorafenib in adult patients with advanced HCC (NCT03606590); it was conducted at six sites across six European countries. The primary endpoint was overall response rate (ORR) compared with historical controls. ORR was defined as the percentage of patients who experienced a response (RECIST version 1.0 for HCC). Secondary efficacy endpoints were infield control rate at 1-year, distant metastases-free survival rate at 1-year, OS at 1-year, and progression-free survival at 6 and 12 months.

The manuscript is well written and pleasant to go through. Although the data lack statistical significance, the information will be of interest to a wide readership, given the patient limitations to design such a study. An important info is lacking, as described below, that is the comparison of adverse effects of TTFields + sorafenib versus historical data on sorafenib alone. Classification of patients on the basis of race/skin color should be deleted because unethical and out of the scope of the study.

  1. Please describe what atezolizumab and bevacizumab are, as well as levatinib.
  2. This phrase needs a reference: Specifically, tumor growth and volume were significantly reduced with TTFields and so-107 rafenib compared with control and either treatment alone (line 107, page 3/17).
  3. Table 1. The term “race” has no scientific background and should be deleted. First, it infringes legislation of some European countries; second, on scientific terms a race is a selected population of livestock obtained for animal production purposes. Also, the authors use skin color to discriminate what they erroneously call “race” and this is unethical, unless the authors justify a relationship between skin pigmentation level (phototypes) and their subject of study. However, it would be probably useful (but in a larger study) to specify the geographical origin of patients at the time of diagnosis.
  4. How compare adverse effects of TTFields + sorafenib versus historical data on sorafenib alone? This information seems important for a phase II study.
  5. The phrase in the discussion (lines 288-290, page 11/17) is inaccurate because Table 2 shows that the comparison with historical controls is not statistically significant: “Data presented here demonstrate that TTFields (150 kHz) concomitant with sorafenib resulted in numerical improvement in outcomes in patients with advanced HCC, as compared to historical controls [6,39-41]”
  6. The phrase in the abstract in lines 45 through 47 is also inaccurate for the same reasons as above. Data should be presented in a factual form, stating that results are not statistically significant (p=0.24).
  7. Author contributions: who wrote the manuscript?

Author Response

Manuscript ID: 1602409

Manuscript title: Tumor Treating Fields Concomitant with Sorafenib in Ad-vanced Hepatocellular Cancer: Results of the HEPANOVA Phase II Study

Manuscript authors: Eleni Gkika, Anca-Ligia Grosu, Teresa Macarulla Mercade, Antonio Cubillo Gracián, Thomas B. Brunner, Michael Schultheiß, Monika Pazgan-Simon, Thomas Seufferlein and Yann Touchefeu

Please note that line numbers to signify the location of changes will be added to this document once the content is final, prior to resubmission

Response to Reviewer 3 Comments

This is an interesting manuscript describing HEPANOVA study,which was was a prospective, open-label, phase II, single arm, historical control study designed to test the preliminary efficacy and safety of TTFields concomitant with sorafenib in adult patients with advanced HCC (NCT03606590); it was conducted at six sites across six European countries. The primary endpoint was overall response rate (ORR) compared with historical controls. ORR was defined as the percentage of patients who experienced a response (RECIST version 1.0 for HCC).Secondary efficacy endpoints were infield control rate at 1-year, distant metastases-free survival rate at 1-year, OS at 1-year, and progression-free survival at 6 and 12 months. The manuscript is well written and pleasant to go through. Although the data lack statistical significance, the information will be of interest to a wide readership, given the patient limitations to design such a study. An important info is lacking, as described below, that is the comparison of adverse effects of TTFields +sorafenib versus historical data on sorafenib alone. Classification of patients on the basis of race/skin color should be deleted because unethical and out of the scope of the study.

Response: The authors thank the reviewer for their suggestions to improve the manuscript.

Point 1: Please describe what atezolizumab and bevacizumab are, aswell as levatinib.

 Response: This information has been included in the introduction.

Point 2: This phrase needs a reference: Specifically, tumor growth and volume were significantly reduced with TTFields and sorafenib compared with control and either treatment alone (line 107, page 3/17).

Response: References have been added, however these will be replaced with details of the preclinical manuscript which was submitted at the same time as this clinical paper

Point 3: Table 1. The term “race” has no scientific background and should be deleted. First, it infringes legislation of some European countries; second, on scientific terms a race is a selected population of livestock obtained for animal production purposes. Also, the authors use skin color to discriminate what they erroneously call “race” and this is unethical, unless the authors justify a relationship between skin pigmentation level (phototypes) and their subject of study. However, it would be probably useful (but in a larger study) to specify the geographical origin of patients at the time of diagnosis.

Response: This information has been removed.

Point 4: How compare adverse effects of TTFields + sorafenib versus historical data on sorafenib alone? This information seems important for a phase II study.

Response: The study was designed so that only the primary endpoint was compared to historical controls. However safety findings are discussed in context with other TTFields and sorafenib studies.

Point 5: The phrase in the discussion (lines 288-290, page 11/17) is inaccurate because Table 2 shows that the comparison with historical controls is not statistically significant: “Data presented here demonstrate that TTFields (150 kHz) concomitant with sorafenib resulted in numerical improvement in outcomes in patients with advanced HCC, as compared to historical controls [6,39-41]”

Response: Additional text has been included to clarify that results were not statistically significant.

Point 6: The phrase in the abstract in lines 45 through 47 is also inaccurate for the same reasons as above. Data should be presented in a factual form, stating that results are not statistically significant (p=024)

Response: The p value has been included, along with text to clarify the results are not significant

Point 7: Author contributions: who wrote the manuscript?

Response: the author contributions have been updated to include critical review and revisions. Medical writing support is already acknowledged in the manuscript

Round 2

Reviewer 3 Report

The manuscript has been improved and is now suitable for publication.